# Optimizing the Surface Structural and Morphological Properties of Silk Thin Films via Ultra-Short Laser Texturing for Creation of Muscle Cell Matrix Model

**DOI:** 10.3390/polym14132584

**Published:** 2022-06-25

**Authors:** Liliya Angelova, Albena Daskalova, Emil Filipov, Xavier Monforte Vila, Janine Tomasch, Georgi Avdeev, Andreas H. Teuschl-Woller, Ivan Buchvarov

**Affiliations:** 1Institute of Electronics, Bulgarian Academy of Sciences, 72 Tzarigradsko Shousse Blvd., 1784 Sofia, Bulgaria; albdaskalova@gmail.com (A.D.); emil.filipov95@gmail.com (E.F.); 2Department Life Science Engineering, University of Applied Sciences Technikum Wien, Höchstädtplatz 6, 1200 Vienna, Austria; monforte@technikum-wien.at (X.M.V.); tomasch@technikum-wien.at (J.T.); teuschl@technikum-wien.at (A.H.T.-W.); 3Austrian Cluster for Tissue Regeneration, 1200 Vienna, Austria; 4Institute of Physical Chemistry, Bulgarian Academy of Sciences, Akad. G. Bonchev Str., 1113 Sofia, Bulgaria; g_avdeev@abv.bg; 5Faculty of Physics, St. Kliment Ohridski University of Sofia, 5 James Bourchier Blvd., 1164 Sofia, Bulgaria; ivan.buchvarov@phys.uni-sofia.bg

**Keywords:** silk fibroin, biopolymers, femtosecond laser processing, muscle tissue engineering, muscle cell matrix 2D model

## Abstract

Temporary scaffolds that mimic the extracellular matrix’s structure and provide a stable substratum for the natural growth of cells are an innovative trend in the field of tissue engineering. The aim of this study is to obtain and design porous 2D fibroin-based cell matrices by femtosecond laser-induced microstructuring for future applications in muscle tissue engineering. Ultra-fast laser treatment is a non-contact method, which generates controlled porosity—the creation of micro/nanostructures on the surface of the biopolymer that can strongly affect cell behavior, while the control over its surface characteristics has the potential of directing the growth of future muscle tissue in the desired direction. The laser structured 2D thin film matrices from silk were characterized by means of SEM, EDX, AFM, FTIR, Micro-Raman, XRD, and 3D-roughness analyses. A WCA evaluation and initial experiments with murine C2C12 myoblasts cells were also performed. The results show that by varying the laser parameters, a different structuring degree can be achieved through the initial lifting and ejection of the material around the area of laser interaction to generate porous channels with varying widths and depths. The proper optimization of the applied laser parameters can significantly improve the bioactive properties of the investigated 2D model of a muscle cell matrix.

## 1. Introduction

Sports injuries, accidents, and other types of muscle trauma can lead to major muscle tears. As a result, the body is not capable of natural endogenous muscle regeneration which may subsequently cause the permanent loss of muscle function and the deterioration of the quality of life of the injured person [1,2,3]. Severe burns, lacerations, or various muscle injuries often require tissue transplantation from either the patient’s own body or from a donor [4,5,6]. Unfortunately, traditional treatment options have many negative consequences for the recipient, such as the creation of a new injury, whose normal healing may be disrupted, leading to a risk of additional infections and a high immune response [7,8,9]. Skeletal muscle tissue engineering, on the other hand, relies on temporary cellular scaffolds that mimic the extracellular matrix (ECM) and provide a stable structure for the natural growth of muscle cells—in certain types of muscle trauma in the body, the matrices can be implanted directly at the site of injury or pre-seeded in vitro with cells and implanted thereafter [10,11,12]. In their elaborate review, Carnes and Pins [13] explain in detail the complex nature of the muscular structure, endogenous regeneration phases, and the advantages and disadvantages of the different muscle tissue engineering approaches. The main purpose of the matrix is to create a biomimetic environment that stimulates cell adhesion, differentiation, and proliferation [14,15,16,17]. In this way, the cells can be reorganized into new three-dimensional tissues. In the process of tissue regeneration, the matrix degrades gradually leaving behind only the newly formed tissue [13,17,18]. Silk fibroin (SF) is one of the most preferred natural polymers for this purpose, as it satisfies all ECM requirements for the creation of a successful temporary cellular scaffold. This is due to SF’s unique mechanical properties, controlled rate of biodegradability, and high biocompatibility [18,19,20,21,22,23,24,25,26,27,28]. All these qualities underlie its wide field of biomedical applications. The most used SF in medical applications is a fibrous protein derived mainly from *Bombyx mori* cocoons [29,30,31,32,33]. It is a fibrous protein showing a high content of the amino acid motif composed of the following aligned amino acid monomers (Gly-Ser-Gly-Ala-Gly-Ala)_n_, which are the molecular basis for its high toughness and strength [32,34,35,36,37]. Detailed information on the bio-applications of SF is given in the comprehensive reviews of Thurber et al. [18] and Holland et al. [26].

In skeletal muscle tissue engineering, fibroin is used mainly in the form of hydrogels or 2D hydro-thin layers [25,32,34,35,38,39]. The creation of “smart” biomimetic muscle tissue matrices based on extracted and purified silk fibroin requires improving their functionality through non-destructive structuring. The functional and physical properties of muscle as a tissue are orientation-dependent qualities [17]—in vivo, ECM structure, characterized by micro grooves between neighboring muscle fibers, guides myoblast alignment during the myotube formation process [13,40,41]. To mimic in vivo muscle organization, different methods have been applied to create biomimetic muscle scaffolds with an aligned structure, including electrospinning [42,43,44,45,46,47], wet and dry spinning [46,48,49,50], and 3D bioprinting [46,51,52,53]. Ultra-short pulse laser treatment is a non-contact, non-invasive, non-destructive, and fully biocompatible method, which generates controlled porosity in biopolymer-based cell matrices [54]—this type of modification leads to the creation of micro and nano structures on the surface of the material that can strongly affect cell adhesion, orientation, and differentiation [48,55]. The method relies on control over the surface characteristics of biomaterials, and accordingly, the growth of future muscle tissue can be directed in the desired direction as microchannels/microgrooves with precisely controlled dimensions, and periodicity can be generated on the scaffold surface in a highly reproducible manner [56,57,58,59,60]. This is very important for muscle tissue engineering, as aligned surface structures are the key to obtaining natural muscle cells’ morphology and orientation [61,62]. The group of Jin et al. [62], for example, achieved uniform laser-ablated microchannels on a substrate that orientated the C2C12 myoblast cells along them, thus helping the natural regenerating process. Apart from that, femtosecond (fs) laser treatment successfully overcomes the limitations associated with the application of other traditional “structuring” methods such as sandblasting or chemical etching that might leave toxic residuals (e.g., from solvents) for the cells in the matrix after treating [32,54]. The side effects (such as microcracking and the absence of molten zones) caused by the interaction of ultra-short laser pulses with biocompatible structures are also minimized [54].

The aim of the presented work is to obtain and design porous 2D fibroin-based cell matrices by femtosecond (fs) laser-induced microstructuring for future application in the engineering of muscle tissue. The surface functionalized samples were characterized by means of morphological (SEM and AFM) and qualitative (EDX, FTIR, micro-Raman, and XRD) analyses, as well as the surface roughness (Sa and Ra) evaluation of the material before and after laser treatment using an optical profilometer was performed. A WCA evaluation, an in vitro degradation test, and initial cellular experiments were also performed.

The analysis of the experimental results clearly shows that femtosecond laser structuring can be applied to assess the surface properties of SF-based cell matrices with a high level of accuracy. By varying the applied fs parameters, different degrees of structuring can be achieved from the initial lifting and ejection of the material around the area of laser interaction to porous channels with different controlled dimensions. Laser modification of the 2D model of muscle cell matrix can significantly improve the bioactive properties of this material, which after the laser parameters’ proper optimization can make its biomedical applications even more successful.

## 2. Materials and Methods

### 2.1. Silk Fibroin Bombyx mori Cocoons Extraction and Samples Preparation

Silk fibroin (SF) was extracted and purified from *Bombyx mori* cocoons (Institute of silkworm breeding, Vratsa, Bulgaria) according to the protocol described in detail in [63]. Briefly, the production of silk fibroin (approximately 7–9% in dH_2_O) consists of degumming with sodium carbonate and lithium bromide (Sigma-Aldrich^®^, Munich, Germany). The procedure includes three main steps: first—preparation of silk cocoons by removing the moth from the cocoon and peeling off the inner layer; second—degumming by boiling the cocoon material in 0.02M Na_2_CO_3_, washing, and drying the degummed silk obtained, a crucial step for sericin (a protein that shields the fibroin in silk fibers) removal, as it is toxic for the cells; and last—dissolving the sericin free silk in 9.3M LiBr solution for 3h at 60 °C. Afterward, the dissolved silk is dialyzed against water for 48h and centrifuged for 10 min at 4618× *g*. The obtained SF (7.26% *w/v* solution) was used for 2D thin layers’ preparation (1 × 1 cm, 110 µm thickness) by spreading the solution on glass slides and removing the prepared thin films samples after drying.

### 2.2. Ultra-Short Laser Texturing of the 2D Fibroin-Based Cell Matrices

The 2D thin layers’ surface microstructuring was performed in air by means of a fs regeneratively amplified Ti:sapphire mode-locked Quantronix-Integra-C system (Hamden, CT, USA), precisely controlled by LabView software. All experiments were performed at λ = 800 nm, ν = 500 Hz, and τ = 150 fs continuous raster surface scanning in XY direction, perpendicular to the SF sample surface, that is positioned on a high-precision XYZ translation stage. The fluence (F) and the scanning velocity (V) were varied as follows F = 0.4 ÷ 2.5 J/cm^2^ and V = 1.7 ÷ 32 mm/s to optimize the dimensions and morphology of the microgrooves created by the laser beam—Table 1, in respect to myoblasts C2C12 cells dimensions, which will be seeded. To promote the natural regeneration process, the distance between the microchannels was also precisely controlled to be optimized with respect to cellular dimensions and orientation inside the channels. According to the literature, widths of grooves and ridges promoting C2C12 alignment and differentiation vary between 20 µm and 100 µm [13,61,64,65]. All analyses of the fs structured samples that followed were averaged on ten separate measurements and performed in respect to the control, a laser non-treated SF scaffold. An illustrative scheme of the experimental setup is given elsewhere [60].

### 2.3. Methods for Characterization of fs Laser-Modified SF Samples

The obtained morphology of the SF 2D thin films after laser processing was investigated by means of Scanning Electron Microscopy (SEM) equipped with an Energy-Dispersive X-ray Spectroscopy module (EDX)—(SEM-TESCAN/LYRA/XMU, Fuveau, France). The samples were gold-sputtered (~ 20 nm Au layer) in vacuum and SEM images were taken at two different magnifications (500× and 3000×/5000×); EDX was performed on an area at higher magnification, the elemental composition was estimated in [wt.%] in respect to control surface. Atomic Force Microscopy (AFM) was also performed. For this purpose, an atomic force microscope MultiMode V (Veeco Instruments Inc., New York, NY, USA) and Controller NanoScope V (Bruker Ltd., Berlin, Germany) in dynamic tapping mode of operation were used. The 2D, 3D, and phase AFM images were taken over an area of 15 × 15 µm^2^ and 5 × 5 µm^2^ via Tap300Al-G (BudgetSensors, Switzerland) silicon AFM probe. Evaluation of samples’ surface roughness profile was additionally performed by a 3D Optical profiler, Zeta-20 (Zeta Instruments, KLA, Milpitas, CA, USA) at 20× magnification. ProfilmOnline software (https://www.profilmonline.com (accessed on 23 March 2022)) was used for better visualization of the 3D true color images obtained; roughness parameters Ra (the mean value of the deviations of the surface height from the median line, according to DIN4776 standards) and Sa (the extension of Ra to a surface area) were also estimated. The samples thickness was measured by a VA 8042 coating meter (Zhejiang, China). In addition to the EDX analysis conducted, the chemical composition of laser treated and untreated surfaces was examined by Fourier-Transform Infrared (FTIR) and micro-Raman Spectroscopy. For this purpose, FTIR spectrophotometer (IR Affinity-1, Shimadzu, Kyoto, Japan), with a working range of 500–4500 cm^−1^, was used for obtaining the IR transmittance spectra [%], and a microRaman spectrometer (LabRAM HR Visible, HORIBA Jobin Yvon, Kyoto, Japan), working with a He-Ne laser (633 nm) and equipped with Olympus BX41 microscope, was used for obtaining the micro-Raman profile of the samples investigated (time of exposition-10s at 100× magnification). For the identification of the crystalline phase of silk fibroin scaffolds, X-ray crystallography analysis was performed within the range of 5–70° θ2 (step size of 0.065° θ2, at continuous scan mode and counting time of 195s) via Philips PW1050 X-ray diffractometer (XRD) system (Philips, Amsterdam, The Netherlands), equipped with a secondary monochromator of the diffraction beam and a copper anode. The phase identification was acquired via QualX2 software through the Crystallography Open Database. Contact Angle (CA) wettability measurements and surface free energy evaluation were performed in air by a video-based optical contact angle measurement device DSA100 Drop Shape Analyzer (KRÜSS GmbH, Hamburg, Germany). For this purpose, two different solutions were used: dH_2_O, and diiodo-methane (DM) in an average volume of 2 µL for a period of 3 min. Contact angles and surface energy were calculated by ADVANCE software (KRÜSS GmbH, Hamburg, Germany) fitting the drop profiles to the Young–Laplace equation and following the Owens–Wendt-Rable–Kaeble (OWRK) equation, respectively. In vitro degradation test performed in PBS buffer saline (pH 7.2, Sigma-Aldrich^®^, Munich, Germany) at 37 °C for a period of two weeks was conducted to predict the stability and biodegradation rate of the fs treated/control SF muscle matrices for when the in vitro cell culture was to be performed. For this purpose, the relative percent weight loss of the scaffolds was calculated at the end of every week and the PBS was replaced with a fresh buffer solution.

Comparison between laser microstructured and untreated SF samples was made in all analyses performed.

### 2.4. Cellular Experiments for Biological Evaluation of Laser-Textured 2D Model of Muscle Cell Matrix

Four groups of fs treated samples were chosen for preliminary cellular experiments: groups G3, G4, G8, and G11 fs treated SF thin films with respect to control G17 (see Table 1 for reference of the fs parameters used). Before seeding the myoblasts cell line C2C12, the samples were sterilized in ethanol for 1 h. Cells were seeded at a density of 5 × 10^4^ cells/cm^2^ in a growth medium (Dulbecco’s modified Eagle’s medium-high Glucose (Life Technologies, Carlsbad, CA, USA), supplemented with 10% fetal calf serum (GE Healthcare, Buckinghamshire, UK), 1% penicillin/streptomycin (Lonza, Basel, Switzerland), and 1% L-glutamine (Lonza, Basel, Switzerland). After 24 h, the medium was replaced by a differentiation medium (Dulbecco’s modified Eagle’s medium-high Glucose (Life Technologies, Carlsbad, CA, USA), supplemented with 3% horse serum (GE Healthcare, Buckinghamshire, United Kingdom), 1% penicillin/streptomycin (Lonza, Basel, Switzerland), and 1% L-glutamine (Lonza, Basel, Switzerland) that was exchanged every second day. Cells were fixed with 4% paraformaldehyde (Roth, Karlsruhe, Germany) for 10 min at room temperature on days 3, 7, and 11 after seeding for analysis of myogenic differentiation by immunofluorescence staining. The staining was performed by washing with dH_2_O and permeabilizing with Tris-Buffered Saline/0.1% (*v/v*) Triton X-100 (TBS/T) for 15 min at room temperature, followed by blocking in PBS/T-1% (*w/v*) bovine serum albumin and 1% (*v/v*) goat serum at room temperature for one hour. The primary antibody targeting all MHC isoforms (MF 20, Developmental Studies Hybridoma Bank, Iowa, USA) was diluted at 1:300 in a blocking solution and incubated overnight at 4 °C. The secondary antibody labeled with Alexa Fluor 488 (Life Technologies, Lofer, Austria) was diluted at 1:400 in a blocking solution and incubated at 37 °C for one hour. Nuclei were labeled by staining with 4′,6-diamidino-2-phenylindole (DAPI) diluted 1:1000 in a blocking solution for 10 min at room temperature. All stainings were analyzed with a Leica DMI 6000b inverted microscope (Leica Microsystems GmbH, Wetzlar, Germany).

The main steps for the preparation and characterization of fs surface functionalized fibroin-based cellular matrices for application in muscle tissue engineering performed in this work are summarized schematically in Figure 1.

## 3. Results and Discussion

### 3.1. SEM, EDX, AFM, and 3D Optical Profiler Analysis of Fs Laser Created Structuredness of SF Based Thin Layers

In Figure 2 contains SEM images of the SF layers, which were fs laser structured while simultaneously varying both the fluence and the scanning velocity in the diapason as follows: F = 0.4 ÷ 2.5 J/cm^2^ and V = 1.7 ÷ 32 mm/s. This process was followed to estimate the optimal laser parameters in order to create structures with the specific dimensions, which were appropriate for cultivating the muscle cells in an oriented manner.

As can be seen from the thickness measurements of the SF thin film samples presented in Table 1 and the selection of the representative morphological SEM images, presented in Figure 2, in all the cases of laser processing an ejection of the material above the basic surface line occurs, which leads to the formation of a thicker, inflated zone in the area of interaction (samples thickness = 115 ÷ 161 µm) in respect to the control group (thickness = 110 µm). At the “gentler” mode of structuring conditions, the laser created zones of interaction, in the form of circular spots, (at V = 32 mm/s and 16 mm/s) which were emerging above the surface baseline (group G1 and group G2), while at a higher applied energy, a material thrown outside of the edges of the created rims was detected, resulting in a hole-like structure (G5, G6, G9, G10, G13, and G14). The basic demand of structuring by the fs laser radiation for obtaining an orientated growth of muscle cells is associated with the formation of groove-like patterns [40,41]. In our experiment, the created grooves limits can be tuned to become narrower and deeper with the increase of F and the decrease of the scanning speed (V), while the highly porous nature of the created microstructures (G3, G4) becomes smoother and more homogenous, a rather granular as opposed to porous morphology. However, material ejection, whether or not it is accompanied by the introduction of additional porosity in the structure, does not lead to damage of the sample’s integrity, nor do cracks or unwanted melting side effects at the groove/spot boundary occur (Figure 2). This fact could be explained by the ultra-fast nature of the processes taking place during fs laser–material interaction, which does not allow for the development of thermal damage effects in the scaffold’s structure, since the interaction ends long before these effects could appear [66,67,68]. In other words, one of the great achievements of ultra-short pulse ablation is the ability to produce a minimal heat-affected zone around the laser spot area. This is because significant accumulated energy is removed during the early stages of material removal and less heat is dissipated beneath the surface. Moreover, in this case, laser processing does not lead to a change in the elemental composition of the samples treated, but only to a slight deviation in the weight concentration [wt.%] of the elements, which can be clearly seen from the results of the EDX analysis performed simultaneously with the SEM (Table 2).

This slight increase in the elemental presence of oxygen [O] in respect to carbon [C] and nitrogen [N] could be explained by the surface oxidation, taking place during the fs laser structuring. Apart from that, the high intensity femtosecond laser–matter interaction, which occurs at higher values of F and lower values of V, leads to the subsequent appearance of O=C–NH bonds’ fragmentation (i.e., a very slight decrease in [C] and [N]), due to the increased material ejection [66].

A representative selection of 3D real-color images, obtained under variation of F and V in respect to a non-treated surface, is given in Figure 3; the corresponding Ra and Sa roughness parameters of all the groups of samples, measured during the optical profilometer analysis, are presented in Table 1. As already mentioned, the specific patterning conditions were chosen in relation to the optimal dimensions and morphology of the patterns created by the laser in respect to the myoblast cells’ suitable seeding conditions [65].

The obtained results are in accordance with the morphological findings acquired from the SEM analysis. As can be seen from Figure 3, the depth and width of the microchannels created by the ultrafast laser, as well as the roughness of the samples (Table 1.), can be varied by tuning the applied laser parameters (F and V). The created grooves have clear cuts with U or V-shaped edges. There is no evidence of mechanical distortion of the biopolymer material. Based on a literature survey, the optimal dimensions for muscle cells vary between 20 µm and 100 µm, as during skeletal muscle formation or regeneration, myoblasts fuse into multi-nucleated tubes to form myofibers, the muscle’s basic “building blocks”, whose diameter ranges in this diapason, depending on the muscle location and function [13,64,65]. By tuning the applied laser parameters (F and V), the SF scaffolds morphology could maximally mimic the ECM of the muscle tissue and be “personally” designed in respect to the specific needs of the seeded cell line. For example, Jin et al. [61], who achieved uniform laser-ablated microgrooves that orientated the C2C12 myoblast cells along them, has varied the spacing between the groove patterns in the range of 0 ÷ 80 µm and have obtained up to 100 µm depth of the channels depending on the energy and number of pulses applied in their experimental work.

The results of the conducted AFM analysis complement those of the SEM and 3D profilometer images and even reveal additional structures at the nano- and micro-levels—nano-roughness, nano- and micro-pores, and sub-microgranulation were observed inside the laser-generated microstructures, which can be clearly seen from the AFM 2D and 3D images of the border zone between the laser-treated and surrounding surface (15 × 15 µm), and the 5 × 5 µm area images inside the laser-generated structures of the SF samples. Some representative AFM images are given in Figure 4.

The AFM images of the control fibroin sample, G17 (Figure 4a), reveal the typical roughness of fibroin films at the nanometric scale. After a laser treatment, the SF films reveal remarkable morphological changes not only at the micro (which is confirmed by SEM and 3D roughness analyses) but also at the nano level: the presence of diverse micro and nanostructures, grains, and pores is clearly observed (Figure 4a,c). Figure 4c visualizes the ejection of the material at the border area of the fs craters created at the four fluences used in this study, but at the highest scanning velocity applied (corresponding to N = 1 in the selection of single-pulse laser mode of operation).

Comparing the data from the performed morphological analysis, no disturbance was observed in the surface integrity at the applied specific conditions of fs laser processing. Optimizing the laser induced micro-features (in respect to the roughness, porosity, and dimensions of the created structures) could subsequently affect muscle cells’ behavior, such as their adhesion, morphology, direction of migration, and differentiation, and hopefully could favor the natural regeneration process of the muscle tissue in vitro, and potentially in vivo [13,69,70,71]. In the last two decades, femtosecond laser processing of different biopolymers for tissue engineering applications have been intensively studied by many research groups; a detailed review of the subject has already been made by Terakawa [54]. Regarding the ultrafast laser structuring of silk fibroin, the information is scarce; there are almost no data on the fs laser modification of the silk protein for bioapplications, nor are there data specifically for muscle tissue engineering. The group of Santos et al. [72], for example, used fs-laser pulses to produce optical waveguides in SF by the direct laser writing of for a biosensor application. In another publication, the same group is further developing their previous results by fs-based printing of well-defined 2D micropatterns of pure and functionalized SF for optical and biomedical applications, such as lab-on-a-chip devices and microsensors [73]. A novel and simple platelet repellent surface was reported by Yang et al., who achieved fabrication of micropattern films based on TA (tannic acid) that could be widely used in the clinical evaluation of antiplatelet therapies [74]. Kim et al., on the other hand, proposed a one-step functionalization of a zwitterionic polymer surface by using a soft lithographic technique [75]. The applied TA-Fe-based coating converted the non-biofouling properties of the polymer to be protein- and diatom-adhesion friendly by a one-step procedure; the lithographic technique provided a regular micropattern for protein and marine diatoms’ surface adhesion.

Based on all the data obtained (in respect to the dimensions and roughness of the microstructures created by the laser processing) and the performed literature survey, the following groups of patterned SF samples were chosen for cellular experiments (in respect to control group 17): F = 0.4 J/cm^2^ and V = 3.8 mm/s (group 3), F = 0.4 J/cm^2^ and V = 1.7 mm/s (group 4), F = 0.8 J/cm^2^ and V = 1.7 mm/s (group 8), and F = 1.7 J/cm^2^ and V = 3.8 mm/s (group 11)—marked in red on the corresponding SEM images of Figure 2 and presented in Figure 3. All fs modifications in the form of individual spots or too “sharp”, narrow, or deep microgrooves created by the laser were excluded as not optimal for directing guided muscle cell growth and the future establishment of functional tissue [13,17,40,41].

The ablation thresholds of the applied fluences were also determined according to the diameter regression technique described in detail in [76]. After the diameter of the craters created on the surface of the SF samples for each scanning velocity used in our study (or the corresponding N-number of pulses) was determined, the corresponding threshold fluences (F_th_) of the material were defined from the plot of squared crater diameters (d^2^) versus the laser fluence for different N (in our case, V as continuous scanning is performed) by extrapolating the curve to zero (Figure 5). According to the logarithmic relationship between D^2^ and F, a linear dependence (well seen from the graph) is evident [76]. Based on this method (by using the equations presented in [76]), for the applied in the current study fluences (F = 0.4 J/cm^2^, F = 0.8 J/cm^2^, F = 1.7 J/cm^2^ and F = 2.5 J/cm^2^), F_th_ were defined as follows: 0.22, 0.18, 0.14, and 0.08 J/cm^2^. As can be seen from the presented graph, the ablation threshold decreases with the decrease of V in the case of continuous scanning (or with increasing N in a single pulse laser mode of operation, respectively). Some representative optical microscope images of laser spots on the SF thin film sample irradiated at the lowest scanning velocity (V = 1.7 mm/s) at every F applied in the current study are also presented in Figure 5; for better visualization of the spot size growth with increasing F at a constant V, the diameter of the spots is also provided.

### 3.2. FTIR, Micro-Raman, and XRD Analysis of SF Scaffolds

Figure 6 summarizes the FTIR transmittance spectra [%] of all the SF laser-treated samples (group 1 ÷ 16) with respect to the control scaffold (group 17).

As can be seen from the presented spectra, there are no deviations in the number, shape, or position of the peaks, while clearly a difference in their intensity is observed with respect to the spectrum taken from the control SF sample. The transmittance spectra exhibit all the characteristic peaks, arising from the peptide bond –CONH–, namely amide I, amide II, and amide III [77,78,79]. All the bands in the FTIR spectra in Figure 6 correspond to C=O stretching (at 1620 cm^−1^) for amide I, N–H bending, and the in-phase combination of C=O bending and C–N stretching (at 1517 cm^−1^ and 1229 cm^−1^, respectively) for amide II and amide III [79]. The decrease in the intensity of the bands representing the data obtained from the laser processed samples can be attributed to the increase of the applied laser energy that causes a disturbance in the vibrations of the amide groups, resulting in a lower peak intensity. This result strongly correlates with the micro-Raman results. The Micro-Raman spectra of the laser-treated matrices (G1 ÷ 16) with respect to the control one (G17) are shown in Figure 7—all the bands characteristic of the amides are well defined, as follows: amide I at 1671 cm^−1^, amide II at 1463 cm^−1^, and amide III at 1274 cm^−1^ [80,81]. The C–H bond at 2945 cm^−1^ and the polarization-dependent peak regarding the Tyr amino acid side-chain at 855 cm^−1^ are also very well pronounced. The polarization-dependent peaks typical for *B. mori* silk at 1401, 1369, 1083, 1001, and 881 cm^−1^ originate from β-sheets formed in the SF structure [82]. The main trend is related to a decrease in the signal intensity after laser treatment, but no change in the number or position of the peaks was observed. Even though some O=C-NH bond fragmentation was detected by the EDX analysis performed on fs processed SF scaffolds (Table 2.), the amide I, amide II, and amide III bands detected in all FTIR transmittance (Figure 6) and micro-Raman spectra (Figure 7) presented are in accordance with the native silk fibroin structure-β-turns (silk I) and β-sheet crystalline silk-II structure, which is a more compact characteristic form of the protein after spinning of the silk fiber by *B. mori* during cocoon formation [81,83].

As a rule, the natural silk fibroin and the degummed SF materials include crystalline and amorphous structures (less stable α-helices, turns, and random coils). The stability of silk fibers is dependent on their β-sheet composition. Crystalline structures have two forms: silk I, a dominant water-soluble helical structural conformation of β-turns, and a water-insoluble silk II structure formed by folded β-sheets [81,82,83,84,85]. The results obtained from the XRD analysis performed on the four groups of laser-processed SF scaffolds, chosen for the preliminary cellular experiments (G3, G4, G8, and G11 laser structured SF thin films in respect to control G17), are given in Figure 8.

As can be seen from the figure, only the XRD spectrum of the G3 SF sample indicated an increased crystallization after laser-induced treatment with respect to the control group (G17) and other fs patterned SF scaffolds—obvious diffraction peaks at 2θ, namely 12.1°, 19.8°, and 24.4° which correspond to the silk I crystalline structure are well pronounced. A lack of well-defined diffraction peaks was observed for silk II in all G3, G4, G8, G11, and G17 XRD spectra (the typical diffraction peaks between 20° and 21°, indicating that the corresponding silk II structures were not detected) [86]. From these findings, it could be concluded that the ultra-short laser processing does not significantly affect the crystal structure of the investigated SF thin films (G4, G8, and G11), as no substantial difference in the XRD spectra is evident when compared with the control SF scaffold (G17). An increased crystallization ability of silk fibroin was observed only after treatment with F = 0.4 J/cm^2^ and V = 1.7 mm/s (G3). Therefore, it is possible that fs laser treatment with the specific parameters leads to the maintenance of silk I’s water-soluble crystalline structure, which could have a positive impact on the protection of the integrity of the fibroin thin films.

### 3.3. Contact Angle Evaluation Analysis

A wettability and total surface energy evaluation (Table 3) of the control (G17) and the laser structured SF thin films, which were chosen for cell studies (G3, G4, G8 and G11), were performed via the sessile drop method using two liquids with different polarities: distilled water (highly polar) and diiodomethane (very low polarity). The obtained results of the Contact angle (CA) evaluation analysis are summarized in Figure 9, where CA change in time is graphically presented; images of water and diiodomethane droplets on 0.00 s and 3 min of application for the SF examined can be also seen in the figure.

As a whole, the results of the both laser-processed and control SF thin films followed a similar trend over the 180s period of wettability evolution: the contact angle decreased in certain boundaries (much more narrow for DM than for dH_2_O), as a slight fluctuation in the total linear behavior was observed at the first 60 s of the droplet contact (for both liquids used) to the fs structured samples (which was not observed on the control SF surface). This could be attributed to a varying amount of entrapped air between the droplet and the surface formed by the laser microstructures, during the liquid’s first contact with the rough surface underneath [87], and this could be explained by the irregular profile of the structures at a submicrometric scale (micro- and nano-pores, grains, etc.) and by a transition between the Cassie–Baxter and Wenzel wetting states [88]. The hydrophilic nature of the scaffolds, attributed to the hydrophilic carboxylic and amino groups in the SF structure [89], was additionally enhanced by the laser processing, especially for the G11 group, where an almost superhydrophilic surface was achieved (WCA dropped from ~ 25 to ~ 20° after 3 min of dH_2_O application). In the case of DM CA evaluation, G3 and G4 were characterized with higher CA in respect to the control group G17. An exception in the droplet behavior was observed in the case of G4, where the CA increased even more after 60 sec. of application. The wettability and the total surface energy of G8 were not measured as DM spread over the entire modified surface at the first second of application and it was not possible for the system to measure the CA. As can be seen from the results presented in Table 3, the surface free energy of the laser-treated surface was also enhanced in respect to the control G17 group.

### 3.4. In Vitro Degradation Test of the 2D Model of Muscle Cell Matrix

To evaluate the stability and biodegradability of the samples investigated, their percent weight losses during the in vitro degradation test performed in PBS (2 weeks at 37oC, 1 mL for each sample) was calculated according to: weight loss (%) = [(W_at the beginning_–W_at the end of the week_)/W_at the beginning_] * 100. The results obtained, which are important for evaluating the stability of the scaffolds for diverse cell culture periods, are given in the table below (Table 4) and visualized by SEM images of the SF scaffolds taken at the end of each week (Figure 10). The measured weight loss of the SF matrices indicates a considerably slow degradation for both the laser structured (G3, G4, G8, and G11) and the control samples (G17), which could be attributed to the already mentioned β-sheet structure of the silk fibroin [81,82,83]. This result is in accordance with the FTIR transmittance (Figure 6) and micro-Raman spectra (Figure 7) obtained and with the works of Wang et al. [25], Farokhi et al. [90], and Lee et al., who estimated a weight loss less than 5% for a period of 14 days during an in vitro degradation of silk-fibroin nanofibrous composite samples [91]. The mechanical properties of the cellular scaffold are a key parameter for in vitro and in vivo tissue regeneration. In the case of skeletal muscle injury, the repair phase, as a part of the regeneration process, takes between 1 and 4 weeks (most often around 2 weeks) for functional regeneration of the myotubes to take place [13]. This process is closely related to muscle satellite cells’ alignment as a basic step for their subsequent differentiation into functional muscle tissue [92].

From the results of the in vitro degradation test and the SEM images, which visualized no appreciable change of the fs groove morphology of the G3, G4, G8, and G11 tested samples, it could be estimated that the fs structured and control SF scaffolds would be significantly stable during the cellular experiments performed afterwards. The ability of the microgrooved scaffold to sustain structural integrity is crucial not only for in vitro experiments but even more for maintaining mechanical stability after body implantation.

### 3.5. Differentiation of Myoblasts on Laser Patterned Silk Fibroin Based Scaffolds

Murine C2C12 myoblasts were seeded on the 2D fibroin matrices with fs laser pre-treated surfaces and differentiated for 11 days. The staining of the nuclei confirmed the presence of C2C12 cells on the samples on days 3, 7, and 11 of the culturing (in blue) with no apparent differences between the fs laser-treated and control samples. The myogenic differentiation was evaluated by immunofluorescence staining of the myogenic marker myosin heavy chain (MHC) in respect to the control SF thin film (Figure 11). Starting from day 3, signs of differentiation can be observed, as indicated by the positive staining for MHC and the elongation of cells. Myogenic development progressed further over the culture period, including the fusion of cells to myotubes at later time points. Fs laser treatment influenced the C2C12 morphology and the organization in the differentiating cells (days 7 and 11). They have a more elongated shape when cultivated on samples G4 and G8, while those seeded on G17-control group have a random organization. Furthermore, the myoblasts seeded on sample G4 align along the grooves from the earliest observed time point on.

## 4. Conclusions

The proposed femtosecond laser induced surface modification method via a selection of different combinations of fluence and scanning velocities is an alternative, non-contact approach for the microstructuring of the SF-based 2D muscle matrices model as it can be successfully applied to the enhancement of scaffolds’ surface properties with a high level of accuracy in respect to the specific cell line needs. By precisely combining the applied laser parameters, different degrees of structuring can be achieved, from the initial lifting and ejection of the material around the area of the laser’s interaction to the generation of porous and granular microgrooves with varying dimensions. At the same time, no side effects such as damage of the sample’s integrity, cracks, melting, or unwanted chemical alternations would could be observed due to the absence of thermal side effects. The ultra-short laser texturing did not affect the elemental composition, morphological integrity, or biodegradability of the SF thin layers; moreover, the hydrophilicity and the surface energy of the scaffolds were enhanced. The performed biological evaluation of the muscle cell compatibility of the laser processed SF matrices demonstrated without a doubt that the cells’ orientation and differentiation were achievable. The analysis of the experimental results clearly shows that laser modification of the 2D model of a muscle cell matrix can significantly improve the surface properties of this material, which, after the optimization of laser parameters, can enhance its biomedical applications. The proposed technique is reliable for the establishment of a fs-microgrooved natural muscle environment model. Our next step is the fs structuring of 3D hydrogel scaffolds that will then be implanted into an animal model for an in vivo evaluation of the silk fibroin-based muscle matrix model.

## Figures and Tables

**Figure 1 polymers-14-02584-f001:**
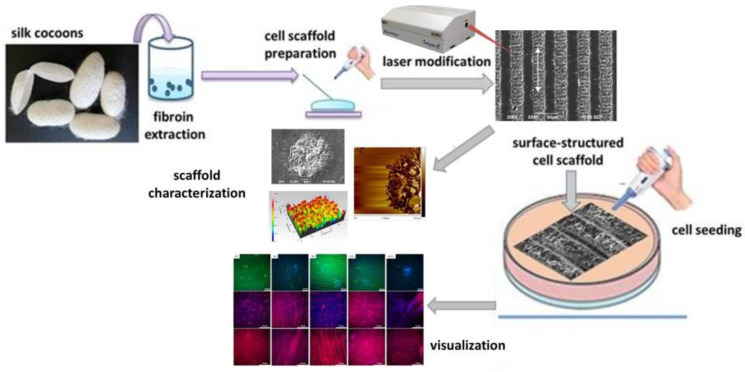
Schematic representation of the main steps for the preparation and characterization of surface-functionalized silk fibroin-based cellular matrices for application in muscle tissue engineering.

**Figure 2 polymers-14-02584-f002:**
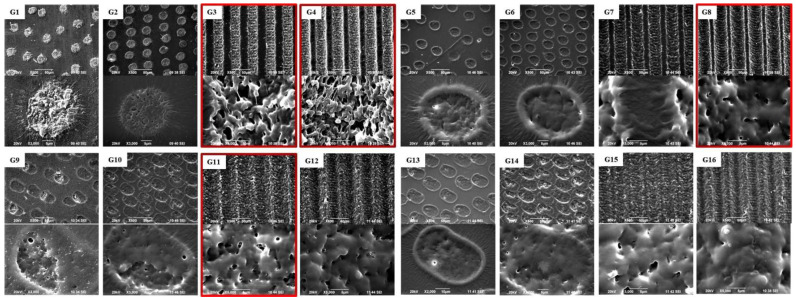
SEM images of G1 ÷ G16 silk fibroin-based cellular scaffolds taken at 500× and 3000×/5000× magnification. In red—SF samples chosen for cellular experiments, based on the results of the analyses performed.

**Figure 3 polymers-14-02584-f003:**
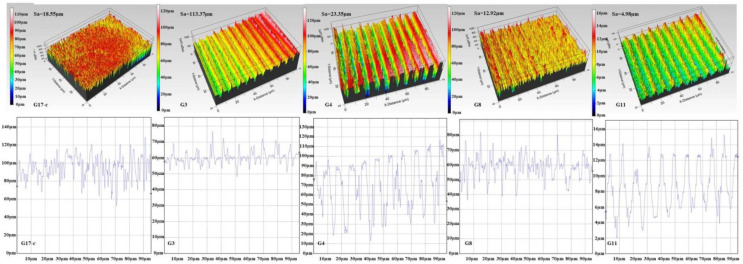
Representative selection of 3D real-color surface profile images of G3, G4, G8, and G11 silk fibroin-based cellular scaffolds in respect to G17-control sample (at 20× magnification); Sa-surface roughness (upper line) and Ra-line roughness cross-section profile (lower line).

**Figure 4 polymers-14-02584-f004:**
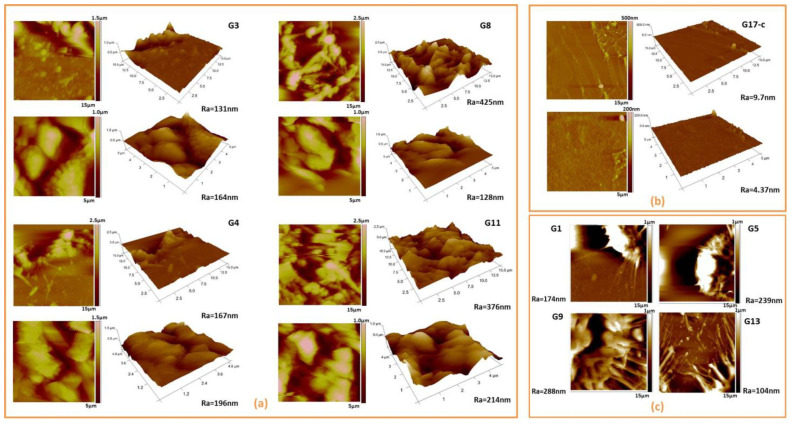
Representative selection of 2D and 3D AFM images at 15 × 15 µm of the border area and 5 × 5 µm inside the laser created structures: (**a**) G3, G4, G8, and G11 silk fibroin-based cellular scaffolds in respect to (**b**) G17-control sample; (**c**) visualizes the ejection of the material at the border area of the fs craters that were created. Local Ra of the areas examined is also given.

**Figure 5 polymers-14-02584-f005:**
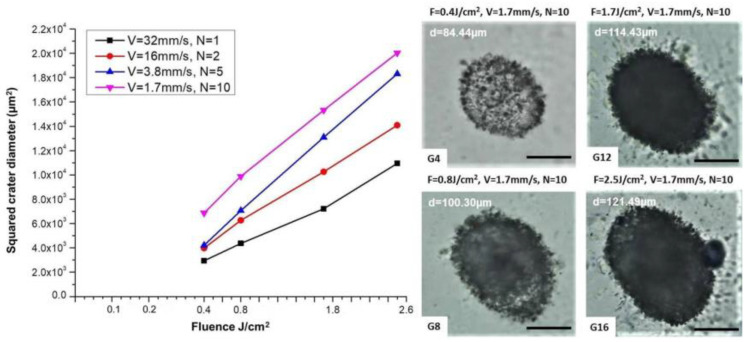
Squared crater diameters (µm^2^) versus the laser fluence for different V (corresponding N) applied on SF sample when irradiated in air (**left**); Optical microscope images of laser spots on SF thin film sample irradiated with N = 10 (at the corresponding lowest scanning velocity V = 1.7 mm/s used) at every F applied in the current study (**right**); scale bar = 50 µm.

**Figure 6 polymers-14-02584-f006:**
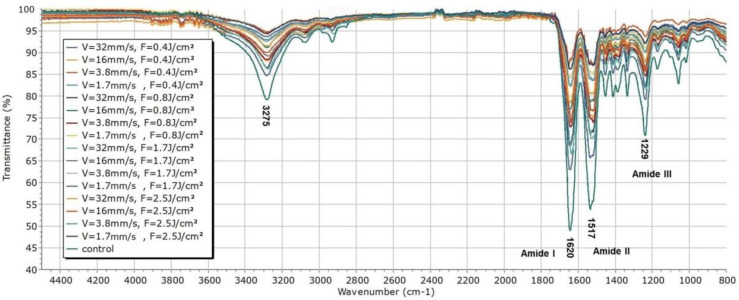
FTIIR Transmittance [%] spectra of G1 ÷ G17 silk fibroin-based cellular scaffolds.

**Figure 7 polymers-14-02584-f007:**
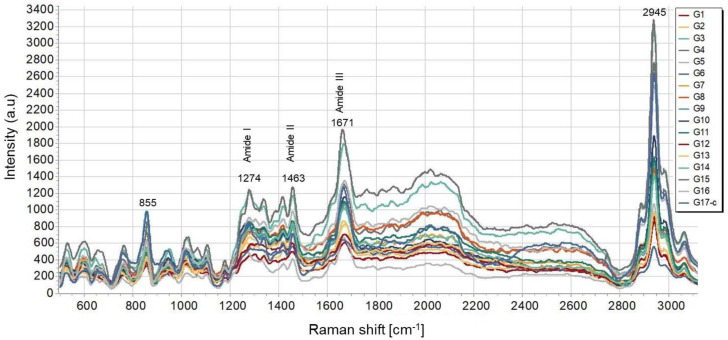
Micro-Raman spectra of G1 ÷ G17 silk fibroin-based cellular scaffolds.

**Figure 8 polymers-14-02584-f008:**
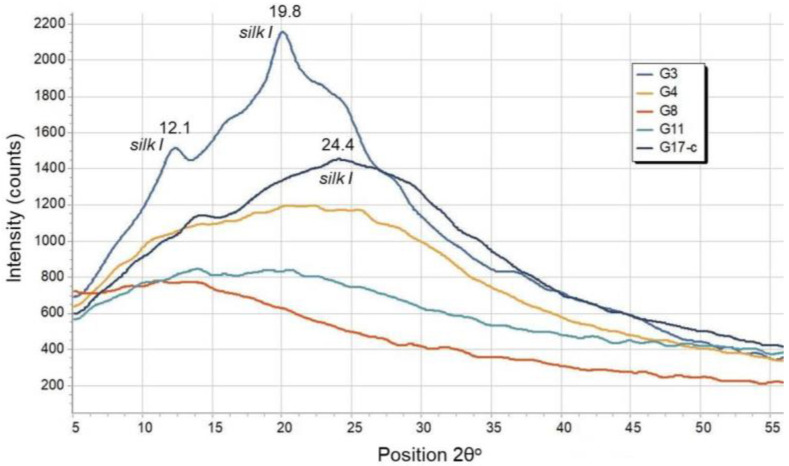
XRD spectra of G3, G4, G8, G11, and G17 SF-scaffolds.

**Figure 9 polymers-14-02584-f009:**
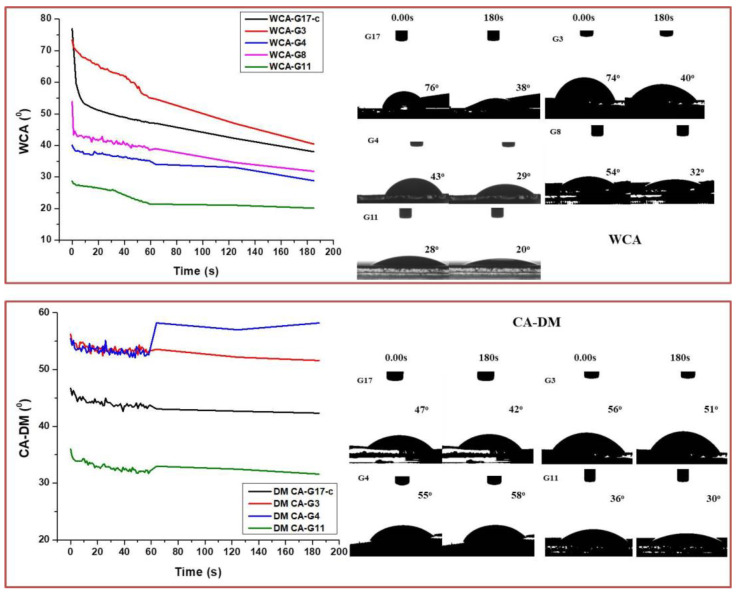
CA evaluation analysis of G3, G4, G8, and G11 silk fibroin-based cellular scaffolds in respect to G17-control sample performed with dH_2_O (**upper** graph) and DM (**lower** graph); images of the droplets taken at 0.00 s and 180 s of application and corresponding CA.

**Figure 10 polymers-14-02584-f010:**
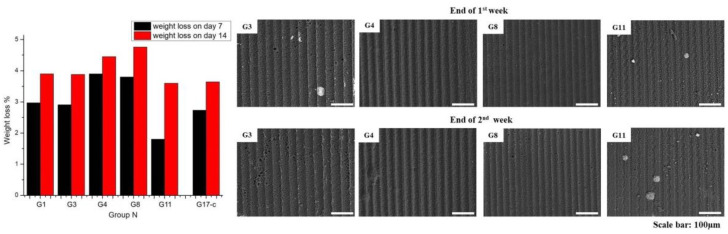
Graphical representation of the weight loss (%) during the 14-day in vitro degradation test and corresponding SEM images of G3, G4, G8, and G11 silk fibroin-based cellular scaffolds at the end of the first and second week of the test. Scale bar = 100 µm.

**Figure 11 polymers-14-02584-f011:**
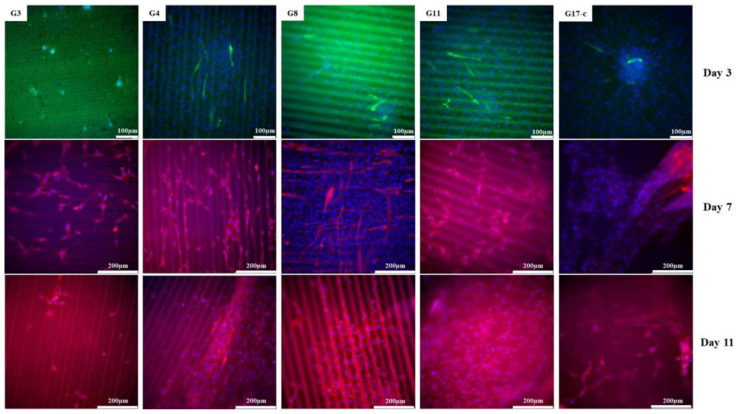
Fluorescence microscopy images (at 20× magnification) of viability (DAPI—nuclei in blue) and muscle differentiation (myosin heavy chain—muscle-specific marker in green on day 3 and in red on days 7 and 11) staining of C2C12 myoblasts cell line, cultured for 3, 7, and 11 days on G3, G4, G8, and G11 fs treated SF thin films in respect to control G17.

**Table 1 polymers-14-02584-t001:** Continuous fs XY raster scan: ƛ = 800 nm, ν = 500 Hz, τ = 150 fs, F = 0.4 ÷ 2.5 J/cm^2^ and V = 1.7 ÷ 32 mm/s, given for each SF sample group (Group No.) treated in respect to control, fs non-treated Group 17. The thickness and the Sa and Ra roughness parameters (in µm), measured for each group of scaffolds are also given.

Group No.	V mm/s	F J/cm^2^	Sa (µm)	Ra (µm)	Thickness (µm)
1	32	0.4	28.11	5.68	115
2	16	0.4	12.2	6.56	121
3	3.8	0.4	113.8	3.74	140
4	1.7	0.4	23.35	12.87	146
5	32	0.8	17.68	0.95	126
6	16	0.8	2.23	1.38	145
7	3.8	0.8	3.18	1.56	149
8	1.7	0.8	12.92	5.79	161
9	32	1.7	12.82	1.53	123
10	16	1.7	7.23	0.64	130
11	3.8	1.7	4.98	0.62	143
12	1.7	1.7	8.02	1.12	156
13	32	2.5	4.89	0.76	134
14	16	2.5	1.52	0.74	137
15	3.8	2.5	2.49	0.62	142
16	1.7	2.5	7.59	1.32	146
17-control	-	-	1.55	0.26	110

**Table 2 polymers-14-02584-t002:** EDX elemental composition given in weight% [wt.%] of each fs treated SF sample group (G1 ÷ G16) in respect to control, fs non-treated Group 17.

EDX Spectrum	C [wt.%]	N [wt.%]	O [wt.%]	Total [wt.%]
G1 V = 32 mm/s, F = 0.4 J/cm^2^	44.87	20.87	34.26	100
G2 V = 16 mm/s, F = 0.4 J/cm^2^	46.08	18.96	34.97	100
G3 V = 3.8 mm/s, F = 0.4 J/cm^2^	45.57	19.16	35.27	100
G4 V = 1.7 mm/s, F = 0.4 J/cm^2^	44.21	21.89	33.9	100
G5 V = 32 mm/s, F = 0.8 J/cm^2^	45.58	20.53	33.89	100
G6 V = 16 mm/s, F = 0.8 J/cm^2^	43.07	23.63	33.3	100
G7 V = 3.8 mm/s, F = 0.8 J/cm^2^	44.03	21.32	34.65	100
G8 V = 1.7 mm/s, F = 0.8 J/cm^2^	42.74	20.23	37.03	100
G9 V = 32 mm/s, F = 1.7 J/cm^2^	46.25	19.79	33.96	100
G10 V = 16 mm/s, F = 1.7 J/cm^2^	43.65	22.92	33.43	100
G11 V = 3.8 mm/s, F = 1.7 J/cm^2^	44.9	21.96	33.14	100
G12 V = 1.7 mm/s, F = 1.7 J/cm^2^	44.59	21.62	33.79	100
G13 V = 32 mm/s, F = 2.5 J/cm^2^	47.96	20.03	32.01	100
G14 V = 16 mm/s, F = 2.5 J/cm^2^	46.52	18.36	35.12	100
G15 V = 3.8 mm/s, F = 2.5 J/cm^2^	48.04	18.01	33.95	100
G16 V = 1.7 mm/s, F = 2.5 J/cm^2^	46.15	19.78	34.07	100
G17-control	48.27	17.95	33.78	100

**Table 3 polymers-14-02584-t003:** Total surface free energy (SFE) evaluation of G3, G4, G8, G11, and G17-c calculated on the OWRK-SFE model based on water and diiodo-methane used as substances. The total SFE of G8 was not obtained, as CA of DM was not possible to measure.

Silk Fibroin Group Sample	Surface Free Energy [mN/m]	Disperse Free Energy [mN/m]	Polar Free Energy [mN/m]
G3	47.89	35.91	11.98
G4	46.91	32.3	14.61
G8	-	-	-
G11	70.92	40.62	30.3
G17-c	37.71	31.85	5.86

**Table 4 polymers-14-02584-t004:** In vitro degradation test in PBS of G3, G3, G4, and G11 and control SF scaffolds (G17) prior to preliminary cellular experiments. Weight loss (%) results on day 7 and day 14 are presented.

Group No.	Weight on Day 1 (mg)	Weight on Day 7 (mg)	Weight on Day 14 (mg)	Weight Loss (%) on Day 7	Weight Loss (%) on Day 14
1	10.1	9.8	9.7	2.97	3.9
3	10.3	10.0	9.9	2.91	3.88
4	10.1	9.7	9.65	3.9	4.45
8	10.5	10.1	10.0	3.8	4.76
11	10.9	10.7	10.5	1.8	3.6
17 control	11.0	10.7	10.6	2.73	3.64

## Data Availability

Not applicable.

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
