# Peer review of "Optimizing the Surface Structural and Morphological Properties of Silk Thin Films via Ultra-Short Laser Texturing for Creation of Muscle Cell Matrix Model"

_polymers, 2022, doi:10.3390/polym14132584_

Round 1

Reviewer 1 Report

This article consisting of the surface conditions of films for their use as biomaterials is very interesting. I really liked the introduction and discussion of results. In addition, the authors have even indicated future prospects for it. For this reason, I recommend its publication in its present form.

Reviewer 2 Report

This manuscript by Liliya Angelova et al. fabricated micropatterned silk films through ultra-short laser for controlled cell differentiation. They characterized the physicochemical properties of the film by SEM, EDX, AFM, FTIR, etc. and further proposed its potential application in muscle tissue engineering. Overall, the authors provided fairly concrete data in the manuscript. This work may be eventually published after addressing the following concerns.

1. Title, ‘Improvement of surface conditions...’, the word ‘conditions’ is too general, I would recommend the authors use another one to make it more specific.

2. It is not necessary to present the Figure 1 in the main text since it is a summary of previous work. This manuscript is a research article, not a review paper.

3. Figure 2, too many details in this illustration, which makes readers confusing. Please re-design.

4. Figure 12, please perform quantitative analysis, rather than just showing these fluorescence images.

5. Some work also related to the facile fabrication of micropattern films for controlled cell attachment could be involved and discussed (Chem. Commun., 2015, 51, 5340; ACS Appl. Mater. Interfaces 2016, 8, 40, 26570).
